# Fixed-Length Poisson MRF:
# Adding Dependencies to the Multinomial

**David I. Inouye**      **Pradeep Ravikumar**      **Inderjit S. Dhillon**
Department of Computer Science
University of Texas at Austin
{dinouye,pradeepr,inderjit}@cs.utexas.edu

## Abstract

We propose a novel distribution that generalizes the Multinomial distribution to enable dependencies between dimensions. Our novel distribution is based on the parametric form of the Poisson MRF model [1] but is fundamentally different because of the domain restriction to a fixed-length vector like in a Multinomial where the number of trials is fixed or known. Thus, we propose the Fixed-Length Poisson MRF (LPMRF) distribution. We develop AIS sampling methods to estimate the likelihood and log partition function (i.e. the log normalizing constant), which was not developed for the Poisson MRF model. In addition, we propose novel mixture and topic models that use LPMRF as a base distribution and discuss the similarities and differences with previous topic models such as the recently proposed Admixture of Poisson MRFs [2]. We show the effectiveness of our LPMRF distribution over Multinomial models by evaluating the test set perplexity on a dataset of abstracts and Wikipedia. Qualitatively, we show that the positive dependencies discovered by LPMRF are interesting and intuitive. Finally, we show that our algorithms are fast and have good scaling (code available online).

## 1   Introduction & Related Work

The Multinomial distribution seems to be a natural distribution for modeling count-valued data such as text documents. Indeed, most topic models such as PLSA [3], LDA [4] and numerous extensions—see [5] for a survey of probabilistic topic models—use the Multinomial as the fundamental base distribution while adding complexity using other latent variables. This is most likely due to the extreme simplicity of Multinomial parameter estimation—simple frequency counts—that is usually smoothed by the simple Dirichlet conjugate prior. In addition, because the Multinomial requires the length of a document to be fixed or pre-specified, usually a Poisson distribution on document length is assumed. This yields a Poisson-Multinomial distribution—which by well-known results is merely an independent Poisson model.[1] However, the Multinomial assumes independence between the words because the Multinomial is merely the sum of independent categorical variables. This restriction does not seem to fit with real-world text. For example, words like "neural" and "network" will tend to co-occur quite frequently together in NIPS papers. Thus, we seek to relax the word independence assumption of the Multinomial.

The Poisson MRF distribution (PMRF) [1] seems to be a potential replacement for the Poisson-Multinomial because it allows some dependencies between words. The Poisson MRF is developed by assuming that every conditional distribution is 1D Poisson. However, the original formulation in [1] only allowed for negative dependencies. Thus, several modifications were proposed in [6] to allow for positive dependencies. One proposal, the Truncated Poisson MRF (TPMRF), simply truncated the PMRF by setting a max count for every word. While this formulation may provide

interesting parameter estimates, a TPMRF with positive dependencies may be almost entirely concentrated at the corners of the joint distribution because of the quadratic term in the log probability (see the bottom left of Fig. 1). In addition, the log partition function of the TPMRF is intractable to estimate even for a small number of dimensions because the sum is over an exponential number of terms.

Thus, we seek a different distribution than a TPMRF that allows positive dependencies but is more appropriately normalized. We observe that the Multinomial is proportional to an independent Poisson model with the domain restricted to a fixed length $L$. Thus, in a similar way, we propose a Fixed-Length Poisson MRF (LPMRF) that is proportional to a PMRF but is restricted to a domain with a fixed vector length—i.e. where $\|\boldsymbol{x}\|_1 = L$. This distribution is quite different from previous PMRF variants because the normalization is very different as will be described in later sections. For a motivating example, in Fig. 1, we show the marginal distributions of the empirical distribution and fitted models using only three words from the Classic3 dataset that contains documents regarding library sciences and aerospace engineering (See Sec. 4). Clearly, real-world text has positive dependencies as evidenced by the empirical marginals of "boundary" and "layer" (i.e. referring to the boundary layer in fluid dynamics) and LPMRF does the best at fitting this empirical distribution. In addition, the log partition function—and hence the likelihood—for LPMRF can be approximated using sampling as described in later sections. Under the PMRF or TPMRF models, both the log partition function and likelihood were computationally intractable to compute exactly.[2] Thus, approximating the log partition function of an LPMRF opens up the door for likelihood-based hyperparameter estimation and model evaluation that was not possible with PMRF.

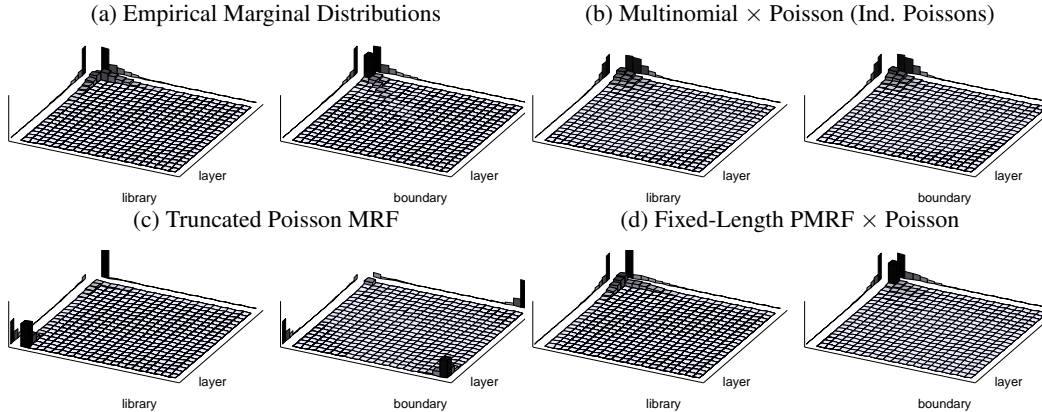

(a) Empirical Marginal Distributions     (b) Multinomial $\times$ Poisson (Ind. Poissons)

(c) Truncated Poisson MRF     (d) Fixed-Length PMRF $\times$ Poisson

Figure 1: **Marginal Distributions from Classic3 Dataset** (Top Left) Empirical Distribution, (Top Right) Estimated Multinomial $\times$ Poisson joint distribution—i.e. independent Poissons, (Bottom Left) Truncated Poisson MRF, (Bottom Right) Fixed-Length PMRF $\times$ Poisson joint distribution. The simple empirical distribution clearly shows a strong dependency between "boundary" and "layer" but strong negative dependency of "boundary" with "library". Clearly, the word-independent Multinomial-Poisson distribution underfits the data. While the Truncated PMRF can model dependencies, it obviously has normalization problems because the normalization is dominated by the edge case. The LPMRF-Poisson distribution much more appropriately fits the empirical data.

In the topic modeling literature, many researchers have realized the issue with using the Multinomial distribution as the base distribution. For example, the interpretability of a Multinomial can be difficult since it only gives an ordering of words. Thus, multiple metrics have been proposed to evaluate topic models based on the perceived dependencies between words within a topic [7, 8, 9, 10]. In particular, [11] showed that the Multinomial assumption was often violated in real world data. In another paper [12], the LDA topic assignments for each word are used to train a separate Ising model—i.e. a Bernoulli MRF—for each topic in a heuristic two-stage procedure. Instead of modeling dependencies *a posteriori*, we formulate a generalization of topic models that allows the LPMRF distribution to directly replace the Multinomial. This allows us to compute a topic model and word dependencies *jointly* under a *unified* model as opposed to the two-stage heuristic procedure in [12].

This model has some connection to the Admixture of Poisson MRFs model (APM) [2], which was the first topic model to consider word dependencies. However, the LPMRF topic model *directly* relaxes the LDA word-independence assumption (i.e. the independent case is the same as LDA) whereas APM is only an *indirect* relaxation of LDA because APM mixes in the exponential family canonical parameter space while LDA mixes in the standard Multinomial parameter space. Another difference with APM is that our proposed LPMRF topic model can actually produce topic assignments for each word similar to LDA with Gibbs sampling [13]. Finally, the LPMRF topic model does not fall into the same generalization of topic models as APM because the instance-specific distribution is not an LPMRF—as described more fully in later sections. The follow up APM paper [14] gives a fast algorithm for estimating the PMRF parameters. We use this algorithm as the basis for estimating the topic LPMRF parameters. For estimating the topic vectors for each document, we give a simple coordinate descent algorithm for estimation of the LPMRF topic model. This estimation of topic vectors can be seen as a direct relaxation of LDA and could even provide a different estimation algorithm for LDA.

## 2 Fixed-Length Poisson MRF

**Notation**   Let $p$, $n$ and $k$ denote the number of words, documents and topics respectively. We will generally use uppercase letters for matrices (e.g. $\Phi, X$), boldface lowercase letters or indices of matrices for column vectors (i.e $\boldsymbol{x}_i, \boldsymbol{\theta}, \Phi_s$) and lowercase letters for scalar values (i.e. $x_i, \theta_s$).

**Poisson MRF Definition**   First, we will briefly describe the Poisson MRF distribution and refer the reader to [1, 6] for more details. A PMRF can be parameterized by a node vector $\boldsymbol{\theta}$ and an edge matrix $\Phi$ whose non-zeros encode the direct dependencies between words: $\text{Pr}_{\text{PMRF}}(\boldsymbol{x} \,|\, \boldsymbol{\theta}, \Phi) = \exp\left(\boldsymbol{\theta}^T \boldsymbol{x} + \boldsymbol{x}^T \Phi \boldsymbol{x} - \sum_{s=1}^p \log(x_s!) - \text{A}\,(\boldsymbol{\theta}, \Phi)\right)$, where $\text{A}\,(\boldsymbol{\theta}, \Phi)$ is the log partition function needed for normalization. Note that without loss of generality, we can assume $\Phi$ is symmetric because it only shows up in the symmetric quadratic term. The conditional distribution of one word given all the others—i.e. $\text{Pr}(x_s | \boldsymbol{x}_{-s})$—is a 1D Poisson distribution with natural parameter $\eta_s = \theta_s + \boldsymbol{x}_{-s}^T \Phi_s$ by construction. One primary issue with the PMRF is that the log partition function (i.e. $\text{A}\,(\boldsymbol{\theta}, \Phi)$) is a log-sum over all vectors in $\mathbb{Z}_+^p$ and thus with even one positive dependency, the log partition function is infinite because of the quadratic term in the formulation. Yang et al. [6] tried to address this issue but, as illustrated in the introduction, their proposed modifications to the PMRF can yield unusual models for real-world data.

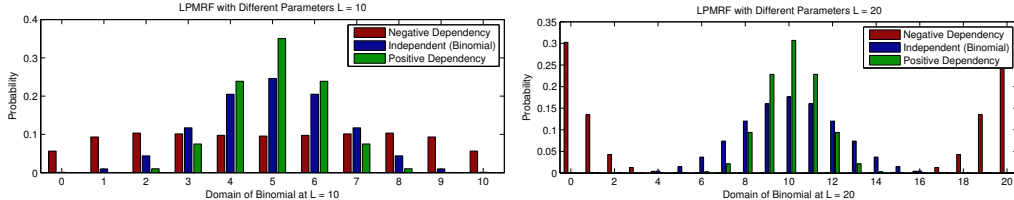

Figure 2: LPMRF distribution for $L = 10$ (left) and $L = 20$ (right) with negative, zero and positive dependencies. The distribution of LPMRF can be quite different than a Multinomial (zero dependency) and thus provides a much more flexible parametric distribution for count data.

**LPMRF Definition**   The Fixed-Length Poisson MRF (LPMRF) distribution is a simple yet fundamentally different distribution than the PMRF. Letting $L \equiv \|\boldsymbol{x}\|_1$ be the length of document, we define the LPMRF distribution as follows:

$$\text{Pr}_{\text{LPMRF}}(\boldsymbol{x}|\boldsymbol{\theta}, \Phi, L) = \exp(\boldsymbol{\theta}^T \boldsymbol{x} + \boldsymbol{x}^T \Phi \boldsymbol{x} - \textstyle\sum_s \log(x_s!) - \text{A}_L(\boldsymbol{\theta}, \Phi)) \quad (1)$$

$$\text{A}_L(\boldsymbol{\theta}, \Phi) = \log \sum_{\boldsymbol{x} \in \mathcal{X}_L} \exp(\boldsymbol{\theta}^T \boldsymbol{x} + \boldsymbol{x}^T \Phi \boldsymbol{x} - \textstyle\sum_s \log(x_s!)) \quad (2)$$

$$\mathcal{X}_L = \{\boldsymbol{x} : \boldsymbol{x} \in \mathbb{Z}_+^p, \|\boldsymbol{x}\|_1 = L\}. \quad (3)$$

The only difference from the PMRF parametric form is the log partition function $\text{A}_L(\boldsymbol{\theta}, \Phi)$ which is conditioned on the set $\mathcal{X}_L$ (unlike the unbounded set for PMRF). This domain restriction is critical to formulating a tractable and reasonable distribution. Combined with a Poisson distribution on vector length $L = \|\boldsymbol{x}\|_1$, the LPMRF distribution can be a much more suitable distribution for documents than a Multinomial. The LPMRF distribution reduces to the standard Multinomial if there are no

dependencies. However, if there are dependencies, then the distribution can be quite different than a Multinomial as illustrated in Fig. 2 for an LPMRF with $p = 2$ and $L$ fixed at either 10 or 20 words. After the original submission, we realized that for $p = 2$ the LPMRF model is the same as the multiplicative binomial generalization in [15]. Thus, the LPMRF model can be seen as a multinomial generalization ($p \geq 2$) of the multiplicative binomial in [15].

**LPMRF Parameter Estimation**  Because the parametric form of the LPMRF model is the same as the form of the PMRF model and we primarily care about finding the correct dependencies, we decide to use the PMRF estimation algorithm described in [14] to estimate $\boldsymbol{\theta}$ and $\Phi$. The algorithm in [14] uses an approximation to the likelihood by using the pseudo-likelihood and performing $\ell_1$ regularized nodewise Poisson regressions. The $\ell_1$ regularization is important both for the sparsity of the dependencies and the computational efficiency of the algorithm. While the PMRF and LPMRF are different distributions, the pseudo-likelihood approximation for estimation provides good results as shown in the results section. We present timing results to show the scalability of this algorithm in Sec. 5. Other parameter estimation methods would be an interesting area of future work.

## 2.1 Likelihood and Log Partition Estimation

Unlike previous work on the PMRF or TPMRF distributions, we develop a tractable approximation to the LPMRF log partition function (Eq. 2) so that we can compute approximate likelihood values. The likelihood of a model can be fundamentally important for hyperparameter optimization and model evaluation.

**LPMRF Annealed Importance Sampling**  First, we develop an LPMRF Gibbs sampler by considering the most common form of Multinomial sampling, namely by taking the sum of a sequence of $L$ Categorical variables. From this intuition, we sample one word at a time while holding all other words fixed. The probability of one word in the sequence $\boldsymbol{w}_\ell$ given all the other words is proportional to $\exp(\boldsymbol{\theta}_s + 2\Phi_s \boldsymbol{x}_{-\ell})$ where $\boldsymbol{x}_{-\ell}$ is the sum of all other words. See the Appendix for the details of Gibbs sampling. Then, we derive an annealed importance sampler [16] using the Gibbs sampling by scaling the $\Phi$ matrix for each successive distribution by the linear sequence starting with 0 and ending with 1 (i.e. $\gamma = 0, \ldots, 1$). Thus, we start with a simple Multinomial sample from $\Pr(x \mid \boldsymbol{\theta}, 0 \cdot \Phi, L) = \Pr_{\text{Mult}}(x \mid \boldsymbol{\theta}, L)$ and then Gibbs sample from each successive distribution $\Pr_{\text{LPMRF}}(x \mid \boldsymbol{\theta}, \gamma\Phi, L)$ updating the sample weight as defined in [16] until we reach the final distribution when $\gamma = 1$. From these weighted samples, we can compute an estimate of the log partition function [16].

**Upper Bound**  Using Hölder's inequality, a simple convex relaxation and the partition function of a Multinomial, an upper bound for the log partition function can be computed: $A_L(\boldsymbol{\theta}, \Phi) \leq L^2 \lambda_{\Phi,1} + L\log(\sum_s \exp\theta_s) - \log(L!)$, where $\lambda_{\Phi,1}$ is the maximum eigenvalue of $\Phi$. See the Appendix for the full derivation. We simplify this upper bound by subtracting $\log(\sum_s \exp\theta_s)$ from $\boldsymbol{\theta}$ (which does not change the distribution) so that the second term becomes 0. Then, neglecting the constant term $-\log(L!)$ that does not interact with the parameters ($\boldsymbol{\theta}, \Phi$), the log partition function is upper bounded by a simple quadratic function w.r.t. $L$.

**Weighting $\Phi$ for Different $L$**  For datasets in which $L$ is observed for every sample but is not uniform—such as document collections, the log partition function will grow quadratically in $L$ if there are any positive dependencies as suggested by the upper bound. This causes long documents to have extremely small likelihood. Thus, we must modify $\Phi$ as $L$ gets larger to counteract this effect. We propose a simple modification that scales the $\Phi$ for each $L$: $\tilde{\Phi}^L = \omega(L)\Phi$. In particular, we propose to use the sigmoidal function using the Log Logistic cumulative distribution function (CDF): $\omega(L) = 1 - \text{LogLogisticCDF}(L \mid \alpha_{\text{LL}}, \beta_{\text{LL}})$. We set the $\beta_{\text{LL}}$ parameter to 2 so that the tail is $O(1/L^2)$ which will eventually cause the upper bound to approach a constant. Letting $\bar{L} = \frac{1}{n} \sum_i L_i$ be the mean instance length, we choose $\alpha_{\text{LL}} = c\bar{L}$ for some small constant $c$. This choice of $\alpha_{\text{LL}}$ helps the weighting function to appropriately scale for corpuses of different average lengths.

**Final Approximation Method for All $L$**  For our experiments, we approximate the log partition function value for all $L$ in the range of the corpus. We use 100 AIS samples for 50 different test values of $L$ linearly spaced between the $0.5\bar{L}$ and $3\bar{L}$ so that we cover both small and large values

of $L$. This gives a total of 5,000 annealed importance samples. We use the quadratic form of the upper bound $U_a(L) = \omega(L)L^2 a$ (ignoring constants with respect to $\Phi$) and find a constant $a$ that upper bounds all 50 estimates: $a_{\max} = \max_L [\omega(L)L^2]^{-1}(\hat{A}_L(\boldsymbol{\theta}, \Phi) - L\log(\sum_s \exp \theta_s) + \log(L!))$, where $\hat{A}_L$ is an AIS estimate of the log partition function for the 50 test values of $L$. This gives a smooth approximation for all $L$ that are greater than or equal to all individual estimates (figure of example approximation in Appendix).

**Mixtures of LPMRF**    With an approximation to the likelihood, we can easily formulate an estimation algorithm for a mixture of LPMRFs using a simple alternating, EM-like procedure. First, given the cluster assignments, the LPMRF parameters can be estimated as explained above. Then, the best cluster assignments can be computed by assigning each instance to the highest likelihood cluster. Extending the LPMRF to topic models requires more careful analysis as described next.

## 3    Generalizing Topic Models using Fixed-Length Distributions

In standard topic models like LDA, the distribution contains a unique topic variable for every word in the corpus. Essentially, this means that every word is actually drawn from a categorical distribution. However, this does not allow us to capture dependencies between words because there is only one word being drawn at a time. Therefore, we need to reformulate LDA in a way that the words from a topic are sampled jointly from a Multinomial. From this reformulation, we can then simply replace the Multinomial with an LPMRF to obtain a topic model with LPMRF as the base distribution. Our reformulation of LDA groups the topic indicator variables for each word into $k$ vectors corresponding to the $k$ different topics. These $k$ "topic indicator" vectors $\boldsymbol{z}^l$ are then assumed to be drawn from a Multinomial with fixed length $L = \|\boldsymbol{z}^j\|$. This grouping of topic vectors yields an equivalent distribution because the topic indicators are exchangeable and independent of one another given the observed word and the document-topic distribution. This leads to the following generalization of topic models in which an observation $\boldsymbol{x}_i$ is the summation of $k$ hidden variables $\boldsymbol{z}_i^j$:

<div align="center">

Generic Topic Model

$\boldsymbol{w}_i \sim \text{SimplexPrior}(\alpha)$

$L_i \sim \text{LengthDistribution}(\bar{L})$

$\boldsymbol{m}_i \sim \text{PartitionDistribution}(\boldsymbol{w}_i, L_i)$

$\boldsymbol{z}_i^j \sim \text{FixedLengthDist}(\boldsymbol{\phi}^j \,;\, \|\boldsymbol{z}_i^j\| = m_i^j)$

$\boldsymbol{x}_i = \sum_{j=1}^k \boldsymbol{z}_i^j$

**Novel LPMRF Topic Model**

$\boldsymbol{w}_i \sim \text{Dirichlet}(\alpha)$

$L_i \sim \text{Poisson}(\lambda = \bar{L})$

$\boldsymbol{m}_i \sim \text{Multinomial}(\boldsymbol{p} = \boldsymbol{w}_i; N = L_i)$

$\boldsymbol{z}_i^j \sim \text{LPMRF}(\boldsymbol{\theta}^j, \Phi^j; L = m_i^j)$

$\boldsymbol{x}_i = \sum_{j=1}^k \boldsymbol{z}_i^j.$

</div>

Note that this generalization of topic models does not require the partition distribution and the fixed-length distribution to be the same. In addition, other distributions could be substituted for the Dirichlet prior distribution on document-topic distributions like the logistic normal prior. Finally, this generalization allows for real-valued topic models for other types of data although exploration of this is outside the scope of this paper.

This generalization is distinctive from the topic model generalization termed "admixtures" in [2]. Admixtures assume that each observation is drawn from an instance-specific base distribution whose parameters are a convex combination of previous parameters. Thus an admixture of LPMRFs could be formulated by assuming that each document, given the document-topic weights $\boldsymbol{w}_i$, is drawn from a LPMRF($\bar{\boldsymbol{\theta}}_i = \sum_j w_{ij} \boldsymbol{\theta}^j, \bar{\Phi}_i = w_{ij}\Phi^j; L = \|\boldsymbol{x}_i\|_1$). Though this may be an interesting model in its own right and useful for further exploration in future work, this is not the same as the above proposed model because the distribution of $\boldsymbol{x}_i$ is not an LPMRF but rather a sum of independent LPMRFs. One case—possibly the only case—where these two generalizations of topic models intersect is when the distribution is a Multinomial (i.e. a LPMRF with $\Phi = 0$). As another distinction from APM, the LPMRF topic model *directly* generalizes LDA because the LPMRF in the above model reduces to a Multinomial if $\Phi = 0$. Fully exploring the differences between this topic model generalization and the admixture generalization are quite interesting but outside the scope of this paper.

With this formulation of LPMRF topic models, we can create a joint optimization problem to solve for the topic matrix $\boldsymbol{Z}_i = [\boldsymbol{z}_i^1, \boldsymbol{z}_i^2, \ldots, \boldsymbol{z}_i^k]$ for each document and to solve for the shared LPMRF

parameters $\boldsymbol{\theta}^{1\ldots k}, \Phi^{1\ldots k}$. The optimization is based on minimizing the negative log posterior:

$$\underset{\boldsymbol{Z}_{1\ldots n}, \boldsymbol{\theta}^{1\ldots k}, \Phi^{1\ldots k}}{\arg\min} \; -\frac{1}{n} \sum_{i=1}^{n} \sum_{j=1}^{k} \Pr_{\text{LPMRF}}(\boldsymbol{z}_i^j | \boldsymbol{\theta}^j, \Phi^j, m_i^j) - \sum_{i=1}^{n} \log(\Pr_{\text{prior}}(m_i^{1\ldots k})) - \sum_{j=1}^{k} \log(\Pr_{\text{prior}}(\boldsymbol{\theta}^j, \Phi^j))$$

$$\text{s.t.} \quad \boldsymbol{Z}_i \mathbf{e} = \boldsymbol{x}_i, \quad \boldsymbol{Z}_i \in \mathbb{Z}_+^{k \times p},$$

where $\mathbf{e}$ is the all ones vector. Notice that the observations $\boldsymbol{x}_i$ only show up in the constraints. The prior distribution on $m_i^{1\ldots k}$ can be related to the Dirichlet distribution as in LDA by taking $\Pr_{\text{prior}}(m_i^{1\ldots k}) = \Pr_{\text{Dir}}(m_i^j / \sum_{\ell} m_i^\ell | \boldsymbol{\alpha})$. Also, notice that the documents are all independent if the LPMRF parameters are known so this optimization can be trivially parallelized.

**Connection to Collapsed Gibbs Sampling** This optimization is very similar to the collapsed Gibbs sampling for LDA [13]. Essentially, the key part to estimating the topic models is estimating the topic indicators for each word in the corpus. The model parameters can then be estimated directly from these topic indicators. In the case of LDA, the Multinomial parameters are trivial to estimate by merely keeping track of counts and thus the parameters can be updated in constant time for every topic resampled. This also suggests that an interesting area of future work would be to understand the connections between collapsed Gibbs sampling and this optimization problem. It may be possible to use this optimization problem to speed up Gibbs sampling convergence or provide a MAP phase after Gibbs sampling to get non-random estimates.

**Estimating Topic Matrices $\boldsymbol{Z}_{1\ldots n}$** For LPMRF topic models, the estimation of the LPMRF parameters given the topic assignments requires solving another complex optimization problem. Thus, we pursue an alternating EM-like scheme as in LPMRF mixtures. First, we estimate LPMRF parameters with the PMRF algorithm from [14], and then we optimize the topic matrix $\boldsymbol{Z}_i \in \mathbb{R}^{p \times k}$ for each document. Because of the constraints on $\boldsymbol{Z}_i$, we pursue a simple dual coordinate descent procedure. We select two coordinates in row $r$ of $\boldsymbol{Z}_i$ and determine if the optimization problem can be improved by moving $a$ words from topic $\ell$ to topic $q$. Thus, we only need to solve a series of simple univariate problems. Each univariate problem only has $x_{is}$ number of possible solutions and thus if the max count of words in a document is bounded by a constant, the univariate subproblems can be solved efficiently. More formally, we are seeking a step size $a$ such that $\widehat{\boldsymbol{Z}}_i = \boldsymbol{Z}_i + a\boldsymbol{e}_r\boldsymbol{e}_\ell^T - a\boldsymbol{e}_r\boldsymbol{e}_q^T$ gives a better optimization value than $\boldsymbol{Z}_i$. If we remove constant terms w.r.t. $a$, we arrive at the following univariate optimization problem (suppressing dependence on $i$ because each of the $n$ subproblems are independent):

$$\underset{-z_r^\ell \le a \le z_r^q}{\arg\min} \; -a[\theta_r^\ell - \theta_r^q + 2\boldsymbol{z}_\ell^T \Phi_r^\ell - 2\boldsymbol{z}_q^T \Phi_r^q] + [\log((z_r^\ell + a)!) + \log((z_r^q - a)!)]$$

$$+ \mathrm{A}_{m^\ell + a}(\boldsymbol{\theta}^\ell, \Phi^\ell) + \mathrm{A}_{m^q + a}(\boldsymbol{\theta}^q, \Phi^q) - \log(\Pr_{\text{prior}}(\tilde{m}^{1\ldots k})),$$

where $\tilde{m}$ is the new distribution of length based on the step size $a$. The first term is the linear and quadratic term from the sufficient statistics. The second term is the change in base measure of a word is moved. The third term is the difference in log partition function if the length of the topic vectors changes. Note that the log partition function can be precomputed so it merely costs a table lookup. The prior also only requires a simple calculation to update. Thus the main computation comes in the inner product $\boldsymbol{z}_\ell^T \Phi_r^\ell$. However, this inner product can be maintained very efficiently and updated efficiently so that it does not significantly affect the running time.

## 4 Perplexity Experiments

We evaluated our novel LPMRF model using perplexity on a held-out test set of documents from a corpus composed of research paper abstracts[3] denoted Classic3 and a collection of Wikipedia documents. The Classic3 dataset has three distinct topic areas: medical (Medline, 1033), library information sciences (CISI, 1460) and aerospace engineering (CRAN, 1400).

**Experimental Setup** We train all the models using a 90% training split of the documents and compute the held-out perplexity on the remaining 10% where perplexity is equal to $\exp(-\mathcal{L}(X^{\text{test}}|\boldsymbol{\theta}^{1\ldots k}, \Phi^{1\ldots k})/N_{\text{test}})$, where $\mathcal{L}$ is the log likelihood and $N_{\text{test}}$ is the total number of words in the test set. We evaluate single, mixture and topic models with both the Multinomial as the base distribution and LPMRF as the base distribution at $k = \{1, 3, 10, 20\}$. The topic indicator matrices $\boldsymbol{Z}_i$ for the test set are estimated by fitting a MAP-based estimate while holding the topic parameters $\boldsymbol{\theta}^{1\ldots k}, \Phi^{1\ldots k}$ fixed.[4] For a single Multinomial or LPMRF, we set the smoothing parameter $\beta$ to $10^{-4}$.[5] We select the LPMRF models using all combinations of 20 log spaced $\lambda$ between 1 and $10^{-3}$, and 5 linearly spaced weighting function constants $c$ between 1 and 2 for the weighting function described in Sec. 2.1. In order to compare our algorithms with LDA, we also provide perplexity results using an LDA Gibbs sampler [13] for MATLAB [6] to estimate the model parameters. For LDA, we used 2000 iterations and optimized the hyperparameters $\alpha$ and $\beta$ using the likelihood of a tuning set. We do not seek to compare with many topic models because many of them use the Multinomial as a base distribution which could be replaced by a LPMRF but rather we simply focus on simple representative models.[7]

**Results** The perplexity results for all models can be seen in Fig. 3. Clearly, a single LPMRF significantly outperforms a single Multinomial on the test dataset both for the Classic3 and Wikipedia datasets. The LPMRF model outperforms the simple Multinomial mixtures and topic models in all cases. This suggests that the LPMRF model could be an interesting replacement for the Multinomial in more complex models. For a small number of topics, LPMRF topic models also outperforms Gibbs sampling LDA but does not perform as well for larger number of topics. This is likely due to the well-developed sampling methods for learning LDA. Exploring the possibility of incorporating sampling into the fitting of the LPMRF topic model is an excellent area of future work. We believe LPMRF shows significant promise for replacing the Multinomial in various probabilistic models.

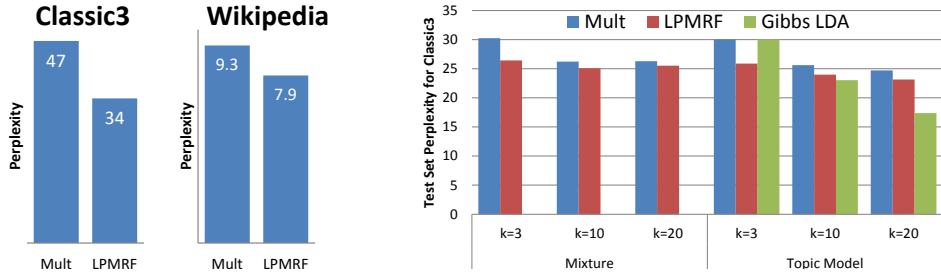

Figure 3: (Left) The LPMRF models quite significantly outperforms the Multinomial for both datasets. (Right) The LPMRF model outperforms the simple Multinomial model in all cases. For a small number of topics, LPMRF topic models also outperforms Gibbs sampling LDA but does not perform as well for larger number of topics.

**Qualitative Analysis of LPMRF Parameters** In addition to perplexity analysis, we present the top words, top positive dependencies and the top negative dependencies for the LPMRF topic model in Table 1. Notice that in LDA, only the top words are available for analysis but an LPMRF topic model can produce intuitive dependencies. For example, the positive dependency "language+natural" is composed of two words that often co-occur in the library sciences but each word independently does not occur very often in comparison to "information" and "library". The positive dependency "stress+reaction" suggests that some of the documents in the Medline dataset likely refer inducing stress on a subject and measuring the reaction. Or in the aerospace topic, the positive dependency "non+linear" suggests that non-linear equations are important in aerospace. Notice that these concepts could not be discovered with a standard Multinomial-based topic model.

Table 1: Top Words and Dependencies for LPMRF Topic Model

| | Topic 1 | | | Topic 2 | | | Topic 3 | |
|---|---|---|---|---|---|---|---|---|
| *Top words* | *Top Pos. Edges* | *Top Neg. Edges* | *Top words* | *Top Pos. Edges* | *Top Neg. Edges* | *Top words* | *Top Pos. Edges* | *Top Neg. Edges* |
| information | states+united | paper-book | patients | term+long | cells-patient | flow | supported+simply | flow-shells |
| library | point+view | libraries-retrieval | cases | positive+negative | patients-animals | pressure | account+taken | number-numbers |
| research | test+tests | library-chemical | normal | cooling+hypothermi | patients-rats | boundary | agreement+good | flow-shell |
| system | primary+secondary | libraries-language | cells | system+central | hormone-protein | results | moment+pitching | wing-hypersonic |
| libraries | recall+precision | system-published | treatment | atmosphere+height | growth-parathyroid | theory | non+linear | solutions-turbulent |
| book | dissemination+sdi | information-citations | children | function+functions | patients-lens | method | lower+upper | mach-reynolds |
| systems | direct+access | information-citation | found | methods+suitable | patients-mice | layer | tunnel+wind | flow-stresses |
| data | language+natural | chemical-document | results | stress+reaction | patients-dogs | given | time+dependent | theoretical-drag |
| use | years+five | library-scientists | blood | low+rates | hormone-tumor | number | level+noise | general-buckling |
| scientific | term+long | library-scientific | disease | case+report | patients-child | presented | purpose+note | made-conducted |

# 5 Timing and Scalability

Finally, we explore the practical performance of our algorithms. In C++, we implemented the three core algorithms: fitting $p$ Poisson regressions, fitting the $n$ topic matrices for each document, and sampling 5,000 AIS samples. The timing for each of these components respectively can be seen in Fig. 4 for the Wikipedia dataset. We set $\lambda = 1$ in the first two experiments which yields roughly 20,000 non-zeros and varied $\lambda$ for the third experiment. Each of the components is trivially parallelized using OpenMP (http://openmp.org/). All timing experiments were conducted on the TACC Maverick system with Intel Xeon E5-2680 v2 Ivy Bridge CPUs (2.80 GHz), 20 CPUs per node, and 12.8 GB memory per CPU (https://www.tacc.utexas.edu/). The scaling is generally linear in the parameters except for fitting topic matrices which is $O(k^2)$. For the AIS sampling, the scaling is linear in the number of non-zeros in $\Phi$ irrespective of $p$. Overall, we believe our implementations provide both good scaling and practical performance (code available online).

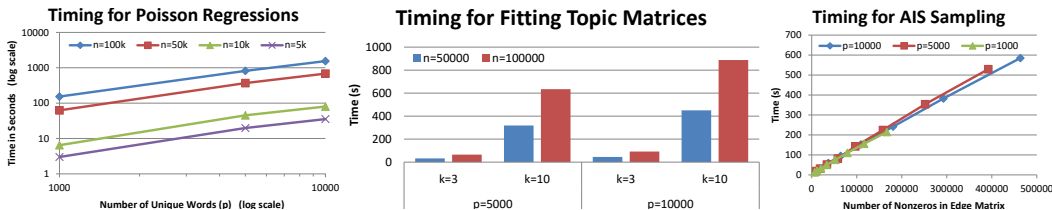

Figure 4: (Left) The timing for fitting $p$ Poisson regressions shows an empirical scaling of $O(np)$. (Middle) The timing for fitting topic matrices empirically shows scaling that is $O(npk^2)$. (Right) The timing for AIS sampling shows that the sampling is approximately linearly scaled with the number of non-zeros in $\Phi$ irrespective of $p$.

# 6 Conclusion

We motivated the need for a more flexible distribution than the Multinomial such as the Poisson MRF. However, the PMRF distribution has several complications due to its normalization that hinder it from being a general-purpose model for count data. We overcome these difficulties by restricting the domain to a fixed length as in a Multinomial while retaining the parametric form of the Poisson MRF. By parameterizing by the length of the document, we can then efficiently compute sampling-based estimates of the log partition function and hence the likelihood—which were not tractable to compute under the PMRF model. We extend the LPMRF distribution to both mixtures and topic models by generalizing topic models using fixed-length distributions and develop parameter estimation methods using dual coordinate descent. We evaluate the perplexity of the proposed LPMRF models on datasets and show that they offer good performance when compared to Multinomial-based models. Finally, we show that our algorithms are fast and have good scaling. Potential new areas could be explored such as the relation between the topic matrix optimization method and Gibbs sampling. It may be possible to develop sampling-based methods for the LPMRF topic model similar to Gibbs sampling for LDA. In general, we suggest that the LPMRF model could open up new avenues of research where the Multinomial distribution is currently used.

### Acknowledgments

This work was supported by NSF (DGE-1110007, IIS-1149803, IIS-1447574, DMS-1264033, CCF-1320746) and ARO (W911NF-12-1-0390).

## Footnotes

[1]The assumption of Poisson document length is not important for most topic models [4].

[2]The example in Fig. 1 was computed by exhaustively computing the log partition function.

[3]http://ir.dcs.gla.ac.uk/resources/test_collections/

[4]For topic models, the likelihood computation is intractable if averaging over all possible $\boldsymbol{Z}_i$. Thus, we use a MAP simplification primarily for computational reasons to compare models without computationally expensive likelihood estimation.

[5]For the LPMRF, this merely means adding $10^{-4}$ to $y$-values of the nodewise Poisson regressions.

[6]http://psiexp.ss.uci.edu/research/programs_data/toolbox.htm

[7]We could not compare to APM [2, 14] because it is not computationally tractable to calculate the likelihood of a test instance in APM, and thus we cannot compute perplexity.

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
