[Supplementary Material]

## A  LPMRF Gibbs Sampling Derivation

As described in the main paper, we develop an LPMRF Gibbs sampler by considering the most common form of Multinomial sampling, namely by taking the sum of a sequence of $L$ Categorical variables. Thus, if $\boldsymbol{w}_1, \boldsymbol{w}_2, \ldots, \boldsymbol{w}_L \sim \text{Categorical}(\boldsymbol{\theta})$ and $\boldsymbol{x} = \sum_{\ell=1}^{L} \boldsymbol{w}_\ell \sim \text{Multinomial}(\boldsymbol{\theta}|L)$, then the probability of any particular sequence is merely the Multinomial probability scaled by the inverse of the multinomial coefficient:

$$\Pr(\boldsymbol{w}_1, \boldsymbol{w}_2, \ldots, \boldsymbol{w}_L \,|\, \boldsymbol{\theta}) = \begin{pmatrix} L \\ x_1, x_2, \ldots, x_p \end{pmatrix}^{-1} \Pr_{\text{Mult}}(\boldsymbol{x} = \sum_{\ell=1}^{L} \boldsymbol{w}_\ell \,|\, \boldsymbol{\theta}, L). \tag{4}$$

In a similar way, we can implicitly derive the probability for a particular sequence of words whose sum is distributed as an LPMRF:

$$\Pr(\boldsymbol{w}_1, \boldsymbol{w}_2, \ldots, \boldsymbol{w}_L \,|\, \boldsymbol{\theta}, \Phi) = \begin{pmatrix} L \\ x_1, x_2, \ldots, x_p \end{pmatrix}^{-1} \Pr_{\text{LPMRF}}(\boldsymbol{x} = \sum_{\ell=1}^{L} \boldsymbol{w}_\ell \,|\, \boldsymbol{\theta}, \Phi, L) \tag{5}$$

$$= \exp(\boldsymbol{\theta}^T \boldsymbol{x} + \boldsymbol{x}^T \Phi \boldsymbol{x} - A_L(\boldsymbol{\theta}, \Phi) - \log(L!)). \tag{6}$$

To develop a Gibbs sampler, we simply need to compute the conditional probability of one of these words given all the other words. Letting $\boldsymbol{x}_{-\ell} \equiv \sum_{m \neq \ell} \boldsymbol{w}_m$, then $\boldsymbol{x} = \boldsymbol{x}_{-\ell} + \boldsymbol{w}_\ell$. Thus, using the fact that the conditional distribution is proportional to the joint distribution, we can derive the form of the conditional distribution:

$$\Pr(\boldsymbol{w}_\ell = \mathbf{e}_s \,|\, \boldsymbol{w}_1, \ldots, \boldsymbol{w}_{\ell-1}, \boldsymbol{w}_{\ell+1}, \ldots, \boldsymbol{w}_L, \boldsymbol{\theta}, \Phi) \tag{7}$$

$$\propto \exp(\boldsymbol{\theta}^T (\boldsymbol{x}_{-\ell} + \boldsymbol{w}_\ell) + (\boldsymbol{x}_{-\ell} + \boldsymbol{w}_\ell)^T \Phi (\boldsymbol{x}_{-\ell} + \boldsymbol{w}_\ell)) \tag{8}$$

$$\propto \exp(\boldsymbol{\theta}_s + 2\Phi_s \boldsymbol{x}_{-\ell}). \tag{9}$$

Thus, each word can be sampled given the state of all the other words thus producing an LPMRF Gibbs sampler.

## B  Derivation of LPMRF Log Partition Upper Bound

$$A_L(\boldsymbol{\theta}, \Phi) \leq \log\Big[ \Big( \sup_{\boldsymbol{x} \in \mathcal{X}_L} \exp(\boldsymbol{x}^T \Phi \boldsymbol{x}) \Big) \sum_{\boldsymbol{x} \in \mathcal{X}_L} \exp(\boldsymbol{\theta}^T \boldsymbol{x} - \sum_s \log(x_s!)) \Big] \tag{10}$$
$$\text{Hölder's Inequality}$$

$$= \log\Big[ \Big( \sup_{\boldsymbol{x} \in \mathcal{X}_L} \exp(\boldsymbol{x}^T \Phi \boldsymbol{x}) \Big) \exp(L \log(\sum_s \exp(\boldsymbol{\theta}_s)) - \log(L!)) \Big] \tag{11}$$
$$\text{Derived from Multinomial}$$

$$\leq \log\Big[ \exp(L^2 \lambda_{\Phi,1}) \exp(L \log(\sum_s \exp \theta_s) - \log(L!)) \Big] \tag{12}$$
$$\text{Convex Relaxation of } \mathcal{X}_L$$

$$= L^2 \lambda_{\Phi,1} + L\log(\sum_s \exp \theta_s) - \log(L!), \tag{13}$$

where $\lambda_{\Phi,1}$ is the maximum eigenvalue of $\Phi$. See next section for derivation of Eqn. 11.

### B.1  Derivation of Multinomial Partition Function

**Lemma 1.** $\sum_{\boldsymbol{x} \in \mathcal{X}_L} \exp(\boldsymbol{\theta}^T \boldsymbol{x} - \sum_{s=1}^{p} \log(x_s!)) = \exp(L \log(\sum_s \exp \theta_s) - \log(L!))$

The derivation is based primarily on the fact that the above expression can be seen as the normalizing factor of a reparameterized Multinomial (or as a scaled version of a standard Multinomial parameterization).

*Proof.*

$$\sum_{\boldsymbol{x} \in \mathcal{X}_L} \exp(\boldsymbol{\theta}^T \boldsymbol{x} - \sum_{s=1}^{p} \log(x_s!))$$

$$= \sum_{\boldsymbol{x} \in \mathcal{X}_L} \exp(\boldsymbol{\theta}^T \boldsymbol{x} - \sum_{s=1}^{p} \log(x_s!) + c\boldsymbol{e}^T \boldsymbol{x} - c\boldsymbol{e}^T \boldsymbol{x} + \log(L!) - \log(L!))$$

(where $\boldsymbol{e} = [1, 1, \ldots, 1]^T$ and $c$ is a constant)

$$= \frac{1}{L!} \sum_{\boldsymbol{x} \in \mathcal{X}_L} \exp((\boldsymbol{\theta} - c)^T \boldsymbol{x} - \sum_{s=1}^{p} \log(x_s!) + c\boldsymbol{e}^T \boldsymbol{x} + \log(L!))$$

$$= \frac{1}{L!} \sum_{\boldsymbol{x} \in \mathcal{X}_L} \exp((\boldsymbol{\theta} - c)^T \boldsymbol{x} - \sum_{s=1}^{p} \log(x_s!) + Lc + \log(L!))$$

$$= \frac{1}{L!} \exp(cL) \sum_{\boldsymbol{x} \in \mathcal{X}_L} \exp((\boldsymbol{\theta} - c)^T \boldsymbol{x} - \sum_{s=1}^{p} \log(x_s!) + \log(L!))$$

$$= \exp(Lc - \log(L!)) \sum_{\boldsymbol{x} \in \mathcal{X}_L} \exp((\boldsymbol{\theta} - c)^T \boldsymbol{x} - \sum_{s=1}^{p} \log(x_s!) + \log(L!))$$

(Letting $c = \log\big(\sum_{s}^{p} \exp(\theta_s)\big)$)

$$= \exp(L\log\big(\sum_{s}^{p} \exp(\theta_s)\big) - \log(L!)) \sum_{\boldsymbol{x} \in \mathcal{X}_L} \exp((\boldsymbol{\theta} - \log\big(\sum_{s}^{p} \exp(\theta_s)\big))^T \boldsymbol{x} - \sum_{s=1}^{p} \log(x_s!) + \log(L!))$$

$$= \exp(L\log\big(\sum_{s}^{p} \exp(\theta_s)\big) - \log(L!)) \underbrace{\sum_{\boldsymbol{x} \in \mathcal{X}_L} \exp(\log(\boldsymbol{\rho})^T \boldsymbol{x} - \sum_{s=1}^{p} \log(x_s!) + \log(L!))}_{\text{Partition function of Multinomial} = 1} \quad (14)$$

$$= \exp(L\log\big(\sum_{s}^{p} \exp(\theta_s)\big) - \log(L!)) \underbrace{\sum_{\boldsymbol{x} \in \mathcal{X}_L} \frac{L!}{\prod_{s=1}^{p} x_s!} \prod_{s=1}^{p} \rho_s^{x_s}}_{\text{Partition function of Multinomial} = 1}$$

$$= \exp(L\log\big(\sum_{s}^{p} \exp(\theta_s)\big) - \log(L!))$$

where 14 is derived by showing that $\boldsymbol{\rho} = \exp(\boldsymbol{\theta} - \log\big(\sum_{s}^{p} \exp(\theta_s)\big))$ is a valid standard Multinomial parameter vector because the vector is positive and sums to 1:

$$\sum_{t=1}^{p} \exp\left(\theta_t - \log\big(\sum_{s}^{p} \exp(\theta_s)\big)\right) = \frac{1}{\sum_{s=1}^{p} \exp(\theta_s)} \sum_{t=1}^{p} \exp(\theta_t) = \frac{\sum_{t=1}^{p} \exp(\theta_t)}{\sum_{s=1}^{p} \exp(\theta_s)} = 1.$$

$\square$

# C   Example of Log Partition Estimation

Figure 5: Example of log partition estimation for all values of $L$.