[Reviews · NeurIPS 2015]

Submitted by Assigned_Reviewer_1

The paper proposes the fixed length PRMs distribution, a distribution that induces dependencies between the dimensions of a multinomial distribution.

The paper is well written and tackles and interesting problem.

Actually, poisson graphical models are currently receiving a lot of attention and help to develop novel forms of probabilistic topic models.

The main goal is to keep the partition function finite. In the present paper, the idea is to restrict the support of the partition function

to the vectors of the same total length/norm.

I like this idea, although the paper would be much stronger if it were comparing to a truncated Poisson LDA, i.e., using TPMRF as presented in [6]. So far, there is only a theoretical discussion of the differences that are not backed up by any empirical illustration.

Related to this, there should be a comparison to the PMRF topic model as presented in [2]. This comparison should show case the benefits of the presented approach to the more general approach of [2]. While I agree that [2] does not feature easily perplexity evaluations because of the tricky partition function, one could still apply some naive MCMC chain on the local Poisson distribution estimated.

At least a comparison as presented in Table 1 should be presented,

to see any benefit in the discovered dependencies.

Furthermore, given that the goal is to estimate a "multinomial with dependencies among the states", the authors should comment

on

just using a "factorisation" of the multinomial distribution into k binary random variables with corresponding dependencies, e.g.,

a tree like dependency or a dependency network?

To summarise, an interesting idea that should explore the connection to the related work more for justification. The benefits over existing approaches are not clearly presented.
Summary: + novel Poisson MRF with an application to LDA + it is very refreshing to revisit the multinomial approach underlying most of machine learning

- baselines should be extended

Submitted by Assigned_Reviewer_2

In order to model positive correlations in discrete data that cannot be handled by ordinary Multinomial distributions, in this paper authors propose a Poisson Markov random fields to directly model word correlations in text data. Specifically, because the number of words in a document is known for textual analysis, they show that the normalizing constant of Poisson MRF can be estimated efficiently, which enables the use of the proposed model in

combination with standard topic models like LDA.

While the assumption is simple and the associated inference is efficient, my concern about this paper is the scalability to ordinary texts. Authors used very small set of texts containing just a thousand or so of documents and thus the dimension of the lexicon is quite small. However, actual texts use a lexicon of over some tens of thousands of words, which will preclude the approach to directly modeling word correlations because the number of parameters in Poisson MRF scales in order O(L^2) with L equals the number of words in lexicon. In fact, ordinary texts do not obey the distribution they assume: the length of a document generally distributes according to a Gamma distribution or like, authors argue that it obeys a Poisson or Normal. When the parameter of a Poisson becomes large, its distribution is quite narrow which is by no means similar to actual distributions.

Therefore, if authors would like to argue that this model is useful,

experiments should be conducted not only the very small data of text data but other types of discrete data that have lower number of dimensions. I think this model is more useful for these domains other than for natural

languages. To model positive and negative word correlations, indirect construction using word embeddings might be much more suitable approach, and in fact pursued in the latest work in [1] at ACL 2015. The direct Poisson MRF approach would be persuasive only when compared with these more appropriate baselines.

[1] "Gaussian LDA for Topic Model with Word Embeddings",

Rajarshi Das, Manzil Zaheer and Chris Dyer, ACL 2015.
Summary: Poisson MRF for directly modeling word correlations, with efficient inference in the normalizing constant. Not scalable for real texts other than very small experiments in this paper.

Submitted by Assigned_Reviewer_3

This paper presents yet another attempt at designing a probabilistic model able to explicitely capture word cooccurences. The authors develop a variant of the Poisson MRF model, where the PMRF densitiy is crucially conditionned on the length - this means that the partition function does not need to sum over all word vectors for all possible lengths; instead the summations run over all documents of the same length. In text modeling, this is a reasonable modeling assumption, since the length is observed anyway. For

this model, the authors show how to approximate efficiently the log partition, using sampling techniques. This model is then used for topic modeling, where the author show how to compute the hidden text topic assignments, and experimentally compared the method with LDA.

This is an enjoyable paper, containing a nice idea: the results presented for the LPMRF indeed show that it can model dependency between pairs of words. It is at the same time a bit frustrating, as details regarding the estimation of the topic model are missing, making the paper more difficult to follow than it should. Furthermore, as is the case with new count models, one would like to see more analyses regarding the fit with actual word counts before jumping to more fancy applications. One other result I would have liked to see is the ability of the model to actually recover in an unsupervised manner the original 3-way partition of Classic3.
Summary: Yet another attempt to model word coccurrences in topic models based on a variant of the Poisson MRT; the lack of details, and the unsufficient thoroughness of the experimental sections are however problematic.

Submitted by Assigned_Reviewer_4

This paper proposes a tractable extension of the Poisson MRF model that can accommodate positive dependencies. This model, the Fixed-Length Poisson MRF (LPMRF), is then used as the emission distribution for an admixture model similar to LDA. In this model, each topic carries additional information about pairwise dependencies (correlations) between word types. Applying this model to text, the authors obtain interesting and competitive results using only 3 topics.

My primary technical question for this work is: Where is the line between modeling topics and modeling word dependencies? If we had a perfect model of word dependencies, would we need a mixture model over topics? Are the topics simply accounting for higher-order (ternary, etc) dependencies that cannot be captured easily by this particular model of word dependencies? With enough topics, the top words are more than enough to look at as results from LDA. The intuitive nature of word type dependencies might be much more questionable if you used your model with more topics. This is especially true because each topic is capturing certain positive and negative dependencies which might interact across topics in ways that are very hard to analyze or present intuitively.

A key concern is scalability of the approach. How well can this method scale with vocabulary size? Especially as the number of topics increases, it seems like the memory requirements might be prohibitive. How many word types were in your data set? Why didn't you test your model with more topics?

This paper is very interesting in terms of its modeling contribution and the development of a fixed-length PMRF with different properties from the original PMRF distribution. The paper has a relatively small contribution in terms of inference, because existing techniques which optimize a pseudo-likelihood are applied. Why did you choose to optimize rather than sample your parameters?

You mention the Admixture of Poisson MRFs model several times in your paper, but you do not discuss any details. Please address this in a related work discussion. In addition, please cite collocation literature from linguistics and contrast your approach; you look for long-range dependences rather than words following one another .The output of the model looks similar, except the order of words in the pairs don't matter.

In section 3, you say that LDA

does not "capture dependencies between words because there is only one word being drawn at a time". While this is true from a generative point of view, I would argue that from an inferential point of view, allowing enough topics helps us to capture many types of positive dependencies.

The proposed model shows only small gains over LDA in terms of perplexity. Perplexity is known to be a bad indicator of topic quality, and there are many existing topic models with better perplexity scores than LDA.

The negative dependencies in Table 1 don't make nearly as much sense as the positive dependences. Please comment on this.

More interesting applications for your model might actually be outside of topic modeling, e.g. in event count modeling for political science where the relationships are less well understood and where the relationships themselves are more relevant as direct objects of study.

Small comments

Bring Figure 1 up to page 2. Figure 1 is very effective. What L did you use to produce these graphs?

Standardize your model names and acronyms. What does CPMRF stand for in Eq 1?

What size of dependency matrix did you use to produce Figure 3?

For perplexity results, Figure 4, note in the caption that lower is better.

Increase your font size in all figures!

Do not use \theta and \Theta as different parameters! Likewise, don't use 'e' both as the well-known constant and as a vector of all 1's in your proof. These notations are hard to follow.

Move your Monte Carlo sampling section before your Upper Bound section in 2.1.

On Page 5, use bold font for "LPMRF" in the generative process to emphasize that this model is your paper's contribution.

In section 4 you say you optimize the Dirichlet parameter \beta. Do you mean \alpha?
Summary: This paper proposes a tractable extension of the Poisson MRF model that can accommodate positive dependencies, and uses this new distribution as a topic-specific distribution over tokens that takes positive and negative pairwise correlations between word types into account. The paper is interesting from a purely statistical point of view, and shows some interesting, yet very incremental, advances in the area of document topic modeling.

Submitted by Assigned_Reviewer_5

This paper proposes a refined family of distributions called the fixed-Length Poisson MRF (LPMRF) which overcomes the intractability of computing log partition function in the original Poisson MRF (PMRF). The core idea is to limit the domain of possible realizations only to the counting vectors whose component sum is fixed a priori. Since LPMRF is discrete distribution, this restriction effectively reduces the event spaces into a finite/bounded set, allowing users to estimate the log partition function via a simple Monte Carlo Sampling. The authors also show an upper bound of the log partition function is linear to the given fixed length. The performance of linear approximation is validated empirically. The application is to generalize topic model so that it considers pairwise dependencies between pairs of words. Instead of sampling each word from the underlying topic independently, the LPMRF topic model samples multiple words as a vector per topic for each document considering the word distributions and the pairwise dependencies of that topic. Due to the exchangeability, the bag-of-words vector for the document could be seen as a simple sum of these indicator vectors, generalizing LDA while maintaining the same basic structure.

As real data in many cases have fixed finite length, LPMRF could be a tractable generalization that can consider arbitrary pairwise word-dependencies beyond n-gram topic models or different from the correlated topic models. The main contribution is two-fold. One is the refined model, and the other is the lower bound of log partition function with its linear approximation. The paper could be further improved if the authors include more analysis about linear approximation such as using more points around the mean as a least square or a regression sense. The main difficulty is to figure out how hard to solve the optimization given in the page 6. It seems the current method repeats EM-fashion algorithm that first estimates LPMRF parameters with the PMRF algorithm and then solve another optimization problem finding the topic assignments given the fixed parameters. It is unclear that using PMRF algorithm is the proper choice for LPMRF setting. Moreover, the second part is solved by finding best portions to move words assigned for a topic to another topics. It is also unclear that this dual coordinate ascent manner is scalable and effective enough for the general number of topics. If these optimizations are more carefully handled (possibly combining with collapsed Gibbs sampling), the paper will be further improved.

For the experimental setting, the current evaluation is minimal. LPMRF is tested only with extremely small number of topics (possibly because its limited scalability in inference). Considering the overwhelming number of parameters comparing to LDA, it is not that surprising LPMRFs with only one or three topics achieve comparable perplexities to the LDA with more than 20 topics. The more interesting part is to compare inter topic qualities: how well topics are separated and how well inter-topic interactions are captured via word-dependencies.

Summary: An interesting and useful model which overcomes the intractability of PMRF is proposed. The inference of solving optimization problems seems not sufficiently explored, and the evaluation is too minimal.

Submitted by Assigned_Reviewer_6

Multinomial distribution is widely used in topic modeling and other applications in machine learning community, due to its simplicity and conjugate property. This paper proposes a novel distribution using a fixed-length Poisson Markov random field. With this new distribution, the authors can capture the dependencies among words in a topic model.

Topic model is a good application for this distribution as dependencies among words are needed for better interpretation. I think the authors motivated the work well and make the paper easy to follow. The results shows the improved perplexity and positive edges captured many common phrases (some are not clear: tests+test)

Some simple analysis on the number of parameters and learning curve would be interesting to help reader understand the complexity of this model.
Summary: This paper proposes a novel distribution, which assigns probability as Multinomial to a fixed-length vector, but can model the dependences among items. The paper is well motivated with topic modeling application and experiments show the improvement.

Author Feedback
Author rebuttal: We thank the reviewers for their insightful and helpful comments. Given the space constraints, we will focus on the inference details and issues of scalability raised by several reviewers as well as provide a correction to a mistake in the original paper that we discovered after submission.

--- Optimization/Inference Details and Scalability ---
We hope to clarify some optimization/inference details below and discuss scalability of our model and algorithms. Essentially, there are two main steps: (1) Learning LPMRF parameters and (2) Learning word-topic assignments. (Let p=#words, n=#documents, k=#documents)

(1) Learning LPMRF parameters: We compute Poisson regressions with L1 regularization for all p variables. Using a naive proximal gradient descent or coordinate descent approach this would be O(p)*O(#iter*n*p), where #iter can be very large. However, using the Newton-like algorithm from APM [2], the computation is approximately O(p)*O(n*p) because the number of Newton-like iterations is approximately constant. In addition, the algorithm is trivially parallel since all the regressions are independent. Finally, the strongest and most interesting dependencies will likely be in the top q words, where q << p. Thus, if we only fit dependencies on the top q = O(sqrt(p)) words, the algorithm becomes O(n*p) and leave the rest of the words independent. For scaling with k topics, we can simple set q = O(sqrt(p)/k) and this part will scale approximately as O(n*k*p) and should thus scale to larger document collections.

(2) Learning word-topic assignments: Here is a pseudo-code for one document with complexity. Let Lmean=mean length of document, max(x)=maximum per-word count, nnzAvg=nnz(thetaEdge)/p.
for s = nonZero(x) --O(Lmean)
.for j1 = 1:k --O(k)
..for j2 = 1:k --O(k)
...for step = {0,2,...,x_s} --O(max(x))
....ComputeObjStep(step,j1,j2,s) --O(1)
...if(stepBest~=0)
....ExecuteStep(step,j1,j2,s) --O(1)
....MaintainVars() --O(nnzAvg)

This gives an overall complexity of O(Lmean * k^2 * max(x) * nnzAvg). Note that Lmean and max(x) can usually be bounded by constants. With these assumptions, the complexity is O(k^2 * nnzAvg). Note that if the model is sparse, usually nnzAvg is O(1) or some small growing function of p. Thus, for fitting all n documents the computational complexity is approximately O(n*k^2*nnzAvg). Like learning the LPMRFs, this is trivially parallel because each document can be fit independently. While we focused on the modeling aspect in the paper, we argue that this proposed method can scale to larger datasets and will seek to add this scalability discussion to the final paper.

--- Correction to Original Paper ---
After the original submission, we noticed a mistake in the log partition upper bound. Instead of being L*\sigma, the function should be L^2*\sigma. After some further investigation, we realized that due to this mistake, the original results were not approximated appropriately. Thus, we developed an annealed importance sampler (AIS) for LPMRF that much more efficiently computes the log partition function than the simple Monte Carlo sampler in the original submission.

We also discovered that this L^2 term in the log partition function strongly dominates the likelihood for large L (e.g. L = 30). Therefore, we propose a very simple and intuitive generalization of the original model where the model uses a weighting function W(L) to downweight the effect of the quadratic term as L increases (merely, x^T*\Theta*x becomes W(L)*x^T*\Theta*x). Note that W(L) = 1 is the original LPMRF model. Using the upper bound as motivation, we suggest that sigmoid-like functions would be a good class of weighting functions where the weight is near 1 for small L and approaching 0 for large L. In particular, we suggest W(L) = 1-LogLogisticCDF(alpha,beta) because the log-logistic distribution CDF is a sigmoid-like function and defined only on non-negative numbers. Furthermore, the log logistic beta parameter controls the rate of decay of the asymptotic tail such that W(L) approaches O(1/L^beta). To asymptotically control the upper bound to be constant, we set beta=2. We select alpha = O(1/\sigma^beta) so that W(L)*L^2*\sigma approaches a constant irrespective of \sigma (i.e. irrespective of the regularization) and select the final alpha based on best training likelihood. We would like to note that this generalized class of weighted length models opens up many other possibilities.

With these updates to our model and algorithms, we redid the experiments for the LPMRF models and the main trends are similar to the original paper:
Single [ Mult: 46.7, LPMRF: 32.4 (30.6% reduction) ]
Mixture (k=3) [ Mult: 29.9, LPMRF: 26.2 (12.3% reduction) ]
Topic Model (k=3) [ Mult: 29.7, LPMRF: 25.3 (14.8% reduction) ]
LDAGibbs [ (k=3): 30.0, (k=10): 23.0, (k=20): 17.4, (k=50): 10.3 ]